# The Effects of Micro-Segregation on Isothermal Transformed Nano Bainitic Microstructure and Mechanical Properties in Laser Cladded Coatings

**DOI:** 10.3390/ma13133017

**Published:** 2020-07-06

**Authors:** Yanbing Guo, Zhuguo Li, Liqun Li, Kai Feng

**Affiliations:** 1School of Materials Engineering, Shanghai Dianji University, Shanghai 201306, China; 2Shanghai Key Lab of Materials Laser Processing and Modification, School of Materials Science and Engineering, Shanghai Jiao Tong University, Shanghai 200240, China; lizg@sjtu.edu.cn (Z.L.); fengkai@sjtu.edu.cn (K.F.); 3State Key Laboratory of Advanced Welding and Joining, Harbin Institute of Technology, Harbin 150001, China; liliqun@hit.edu.cn

**Keywords:** laser cladding, nanostructured bainite, micro-segregation, retained austenite, stability, mechanical properties

## Abstract

The design of metastable retained austenite is the key issue to obtain nano bainitic steel with high strength and toughness. In this study, nanostructured Fe-based bainitic coatings were fabricated using laser cladding and following isothermal heat treatment. The microstructures and mechanical properties of the laser cladded coating were investigated. The results show that the Mn, Cr, Co, and Al segregated at the solidified prior grain boundaries. The micro-segregation of the solutes strongly influenced the stability of the austenite. As the isothermal temperature decreases, the interface of the bainite and blocky retained austenite approach to the prior interdendritic regions with the decreasing isothermal temperature, and the final volume fraction also decreases. The volume fractions of each phase and microstructure morphology of the coatings were determined by the interdendritic micro-segregation and isothermal temperatures. The stability of the blocky retained austenite distributed at the interdendritic area was lower than that of film and island-like morphology. This phenomenon contributed to the ductile and tough nano bainitic coatings with tunable mechanical properties.

## 1. Introduction

The developing advanced high strength steels (AHSSs) with remarkable strength and ductility balance is stimulated attraction for many industries, in order to reduce the structural weight and improved safety [1,2]. Nano structured bainite steel is one of the new series of AHSSs, which consists of nanoscale bainite ferrite and stable retained austenite based on “incomplete transformation” [3,4,5]. The nano bainite is obtained by improving the content carbon and other elements [6]. With the addition of carbon, the martensite starting temperature (Ms) and bainite starting temperature (Bs) are decreasing, and the bainite can be obtained at low temperatures, i.e., 150–300 °C [6]. Meanwhile, high silicon (>1.5 wt%) content can prevent the carbide precipitation, to obtain the nanoscale bainite ferrite plates and carbon enriched retained austenite [7,8]. The nanoscale microstructure and high density of dislocation is formed during the growth of bainite ferrite, leading to the higher strength (1.6–2.5 GPa) and toughness (30 MPa m^1/2^) [9,10]. The retained austenite between bainite ferrite plates and sheaves is the ductile phase [9]. However, the balance of strength and ductility is determined by the volume fraction of the bainite ferrite and retained austenite (RA). Especially, the stability of the retained austenite is the decisive factor of the mechanical properties due to the transformation-induced plasticity effect [11].

The stability of the retained austenite depends on chemical composition [12,13], grain size [14,15,16], and morphology [17,18]. The carbon content is the main contribution to the retained austenite stability [1,12]. The film RA distributed in the lamellar nano ferrite as a result of carbon diffusion from adjacent ferrite during the bainitic transformation process. The ferrite transformed at the range of bainite transformation temperature is defined as bainite ferrite to distinguish that obtained at higher temperature [19,20]. The Bs and Ms is decreased by the high content of carbon in the bainitic steel alloy, which guarantees the nanoscale bainite ferrite transformation at low temperature and the stability of the retained austenite. However, the extensive application of this special steel with excellent mechanical properties is limited. The multi-step heat treatment [21] and martensite pretransition [22] methods has been successfully applied to accelerate the transformation process, and to increase the complexity process. Another method is the refinement of prior austenite grain size and the addition of small amounts of elements, such as aluminum and cobalt [23]. According to the previous studies of the authors, laser cladding and subsequent isothermal holding (LCFSIH) can be applied to fabricate nanostructured bainite efficiently [24]. The rapid solidification process provides a refined prior austenite grain that can shorten the whole transformation time [25]. Therefore, the LCFSIH method is an uncomplicated route for the fabrication of nano bainitic surface layers with high strength and toughness.

The austenite stabilization alloying elements such as Mn, Cr, Si, etc., always segregated at the interdendritic regions during the solidification process, which will significantly affect the final microstructure and properties of the coatings [26]. The present work intended to study the effect of the solutes micro-segregation and isothermal temperatures on the microstructure of the laser cladded coatings. The effects of the morphology and stability of the RA on the mechanical properties were investigated by the characterization of the element distribution in the dendrites and the carbon evolution during the isothermal process.

## 2. Materials and Experimental Procedures

### 2.1. Experiment Materials

The Fe-based powder (particle size of 75–250 μm) was prepared using plasma rotating electrode process. The mild steel plates were chosen as the substrate materials. The chemical composition of the powder and substrate is listed in Table 1.

The substrate steel was machined to the dimension of 140 × 30 × 20 mm. The surface used for laser cladding was prepared by using abrasive blasting. Then, the substrate specimens were cleaned with acetone. The Fe-based powders were dried in the vacuum oven under 150 °C for 2 h.

### 2.2. Laser Cladding

The 3.5 kW diode laser system (Rofin DL-035Q) (Rofin, Frankfurt, Germany) equipped with a coaxial powder feeding system was used to fabricate the coatings. The wavelength of the laser beam is in a range from 808 nm to 940 nm. The spot size of the laser at the focal length (165 mm) is 3.0 × 4.5 mm. The Fe-based powder was deposited on the substrate using an optimized parameter (laser power and scanning rate). The coatings were sufficiently protected by the coaxial and tailed shielding system with argon gas (99.9%). The substrate was preheated on a temperature control electrothermal board (CTBRAND CT-946A) (CTBRAND, Shantou, China). The preheated temperature (200, 250, and 300 °C) was the same as the following isothermal bainitic transformation temperature. The laser cladded coatings were transferred from the preheated board to furnace (Nabertherm N-17-HR-P300) (Nabertherm, Bremen, Germany) for an isothermal transformation immediately. The surface temperature of the coatings was monitored using a digital thermometer (Anritsu HA400-E) (Anritsu, Kanagawa, Japan), to ensure the temperature was above the setting temperature of the furnace before the transferring. The specimens were quenched in the water (20 °C) after isothermal transformation. The parameters used for laser cladding and isothermal holdings are listed in Table 2 and Table 3, respectively.

### 2.3. Characterization

The microstructure of the coating was observed using OM (Zeiss Axioplan 2) (Zeiss, Jena, Germany) and FE-SEM (JEOL JEM-7600F) (JEOL, Tokyo, Japan) after etching in 4% nital (4 vol.% nitric acid and 96 vol.% ethanol) (Sinopharm, Shanghai, China) and picric acid solutions (for interdendritic observation) (Sinopharm, Shanghai, China). The coatings specimens were machined in the form of 3 mm discs of 70 µm thickness so that they could be thinned and electropolished for TEM. A twin-jet electropolisher (Jiaoda, Shanghai, China) operated with a voltage of 40 V in a electrolyte (5% perchloric acid and 95% ethylalcohol) at −30 °C at 40 V. The specimens were examined in JEOL JEM-2100F (JEOL, Tokyo, Japan) operated at 200 kV.

X-ray experiments were conducted using an Ultima IV X-ray diffractometer (Rigaku, Tokyo, Japan) with a scanning rate of 1 °/min over the range 2θ = 35~105°, and unfiltered Cu Kα radiation. The system was operated at 40 kV and 30 mA. The carbon contents and volume fraction of retained austenite and were calculated by integrated intensities of peaks of austenite and ferrite. Four peaks of each phases were utilized to avoid the bias resulting from any crystallographic texture in specimens. Lattice parameter was evaluated utilizing Cohen’s method [27]. The carbon contents in α and γ were calculated from the lattice parameters obtained by the corresponding diffraction peaks according to Ref [28].

The nanoindentation tests were performed using s CETR-Apex nano-mechanical system (CETR, Santa Clara, USA) equipped with a Berkovich indenter tip. The load of 10 mN with a constant loading rate of 0.33 mN/s and dwell time of 5 s were applied on the polished surface of the coatings. Prior to testing, the tip area function and frame stiffness were calibrated using a fused silica reference sample. Moreover, the distribution of nano-hardness was performed by 25 indentations (a square array of 5 × 5 indentations covering a 20 μm × 20 μm area).

The tensile testing samples were sectioned longitudinally from the laser cladded direction, as shown in Figure 1. The total length is 56 mm. The length and width of the test area are 30 mm and 1.0 mm, respectively, and the thickness is about 0.8 mm (the thickness of the laser cladded coatings). The elongation was measured by an extensometer (MTS, Norwood, USA) with a gauge length of 10 mm. In all the tensile tests, a crosshead speed of 0.05 mm/min was used.

## 3. Results and Discussion

### 3.1. Microstructure

Figure 2a shows the image of the laser cladded coating (cross-section). Defects such as pores and cracks are prevalent in laser-cladded steels [29], especially when the chemical compositions of the substrate and coating are severely different, or the laser cladding parameter is inappropriate. No obviously cracks or pores are observed on the cross-section, which indicates that the defects (solidification cracking, cold cracking and pores) may be reduce or eliminate by using pre-heating of the substrate and optimized laser cladding parameters. A low dilution rate indicates a good metallurgical bonding between the coating and the base metals, while guaranteeing the designed chemical compositions. The microstructures are composed of dendrite and interdendritic areas which are depicted as the white and dark regions in Figure 2b. The magnified view near the interdendritic region depicts that the microstructure of the white area is retained austenite (RA), and the dark regions are of thin bainitic sheaves distributing in the inner dendritic areas, as shown in Figure 2c.

The austenite retained after a long isothermal transformation means the stability of the prior austenite in the interdendritic region is higher than that in the inner dendrite areas. The details of the solidified and isothermal transformed microstructures were characterized by the SEM and EDS, to clarify the reason for the enhanced stability of the austenite, as shown in Figure 3(a),(b). The fluctuation of the EDS line scanning across the interdendritic area in the coating is caused by the micro-segregation of the alloy elements during the laser cladding solidification process. The chromium, manganese, molybdenum, and silicon are positively segregated at the interdendritic region, and the aluminum is negatively segregated by contrast. The segregation of the alloy elements (such as Cr and Mn) not only enhances the stability of the austenite [30,31], but also reduce the stacking fault energy of the RA [32,33].

Figure 4 shows the details of the microstructure of the coatings after isothermal transformation. Two typical microstructures can be observed, which are fine microstructures composed of sheaves in the inner dendrite and coarse inter-dendrite structures. The coarsen inter-dendrite structures distribute at the dendrite boundaries of the columnar solidification structure.

The microstructures of the sheaves are nanostructured bainite, which consists of bainite plates (sub-units) and films like retained austenite interlacing distributed in inner dendrite regions. The distribution of bainite sheaves is clutter, with various growth directions. The bainite plates are short and thin. And the bainite sheaves are hardly visible because the growth directions of bainite plate are different from the neighbor plates. It means that more bainite variants nucleate at a lower transformation temperature, e.g., Figure 4a in 200 °C. The shorter the bainite plate caused by impingement with other precipitates ceases the growth of the bainite plate at a specific value [34]. The morphology of bainite will be altered with isothermal temperatures. The sheaves grow longer and thicker under higher transformation temperature. The bainite sheaf always grows from the boundary until it is stopped by the boundaries of the other side (Figure 4c) in 300 °C.

The coarse interdendritic distributing at prior austenite dendrite boundaries (the white dash line) are retained austenite. As mentioned previously, the element segregation during the solidification will coarsen the retained austenite in the inter dendrite region. Austenite stabilized elements such as Cr, Si, Mn, Mo have a gradient across the inter dendrite region, which has been discussed in the author’s previous publications [25]. The chemical gradient will cause the driving force accompanying the transformation of austenite to ferrite (ΔGγα = Gα-Gγ), which is decreased from the boundary to the center of the inter dendrite region. As shown in Figure 5, a thermal dynamic boundary of the bainitic transformation at a specific temperature will exist near the dendrite boundaries. This critical boundary caused by the stability of austenite will carry forward to the center with the decreasing transformation temperature. Therefore, thinner and scattered RA can be observed in the coating at 200 °C, compared with that thicker and blocky austenite distribute at the 300 °C.

As shown in Figure 6 The microstructures in the coatings after three different isothermal heat treatment processes consist of bainite ferrite, island-like retained austenite and thin-film austenite. The island-like and film-like austenite are formed as a result of the carbon diffusion during the isothermal transformation process, to make them carbon enriched. The island-like retained austenite of the relatively high transformed temperatures are usually larger than that of the low temperatures. Because the carbon has a relatively higher diffusion coefficient and lower carbon content on the T_0_ curves at higher temperature [35]. The carbon of the sheaves make the surrounded island-like austenite quickly be stable, and to stop the bainite nucleate at the adjacent region. The surrounding stable region usually has a larger area at a high transformed temperature. Long and thicker bundle of bainite sheaves distribute averagely in the inner dendrite regions for the coatings transformed at 300 °C. By contrast, the shorter and thinner bainite plates intricately distribute at the coating transformed at 200 °C.

The growth morphology of the relatively lower temperature transformed bainite is influenced by the strength of the austenite and the transformation driving force. Direct observations have shown that there will be considerable plastic relaxation in the austenite adjacent to the bainite plates [36]. The dislocation debris generated in this process resists the advance of the ferrite/austenite interface, and the resistance is larger for stronger austenite. The yield strength of the austenite increased with decreasing temperature [37], then a thinner size of the bainite plate after transformation [36].

Figure 7 shows the typical TEM micrograph of film-like bainite structures in 200 and 300 °C transformed coatings. The film-like structures are on the nano-scale. Electron diffraction patterns show that the film-like structures are composed of α and γ, i.e., bainitic ferrite and retained austenite. The stacking faults can be observed near the interdendritic regions in the lower temperature transformed coatings, as shown in Figure 7a.

Figure 8 shows the TEM image of the stacking fault bundles in the coatings isothermal heat treated at 200 °C. The plan-view of the micrographs were taken near the [110] projection. The dark field image and selected area electron diffraction (SAED) pattern of the stacking fault shows a single fcc phase. The SAED pattern illustrates that the stacking fault is overlapped on (111) planes, and the shear band can be formed. Nano-twins always form when the stacking faults overlap successive on the (111) planes. ε-martensite forms when the stacking faults overlap on alternate (111) planes. High density debris is observed on both sides of the twins in the γ phase. The stacking fault energy is deceased in the interdendritic regions as a result of the segregation of Cr, Mn, and Si. Together with the stain and stress field caused by bainite transformation, stacking fault bundles were formed in RA.

The TEM micrographs of the ε-martensite and its corresponding SAED pattern in 200 °C transformed coating are presented in Figure 9. A hcp structure with a lattice parameter of a = 0.39 nm, c = 0.24 nm is calculated according to the SAED pattern. It illustrates that the nano scale lamellar structure is ε-martensite. The ε-martensite is often observed in the deformed austenite at relative low temperature. The overlapping stacking fault is formed regularly on every second (111) plane. The ε-martensite can be observed in the austenite when nanostructured bainite transformed under deformations [38]. The planar stacking faults defects in structure and high strain values can be induced by laser processing. Therefore, the ε-martensite can be formed by a local higher strain during the laser cladding process. The retained austenite with chemical segregation can displacive transform to the hcp-martensite more easily by Shockley partial dislocations gliding [39]. Meanwhile, the trends for the forming ε-martensite increased with the temperature gradient during the cooling process, which means that the low transformed temperature can form more ε-martensite.

Whenever the shear band appeared, the stress-induced martensite can be transformed by strain inducing [40]. It is known that the formation of the stress-induced martensite is related to the overlapping of the stacking faults on ${\left\{111\right\}}_{\gamma }$ close-packed atom plane [41]. The thermal stress after laser cladding and phase transformation induced high stress-strain fields during the bainitic transformation process. The strain-stress filed induced the stacking faults and nano-twins in the austenite, and stress-induced ε-martensite can be formed during this process [42].

The Md temperature is the critical factors for the transformation of the deformation-induced martensite transformation, which can be calculated by JMatPro7.0 software. The chemical composition near the interdendritic regions has been estimated in the previous publication of the authors [25]. The Md temperature of interdendritic regions is estimated to be 145.7 °C, which means that the ε-martensite in the retained austenite is formed during the cooling process. When the austenite with a high stress-strain field cooling to the temperature between Md and Ms, the formation of stacking faults and ε-martensite were promoted. The stacking fault energy is relatively lower for a partial strain-induced or stress-assisted transformation of the austenite into $\alpha^{\prime }$ or ε-martensite [43]. This small amount of the martensite is critical for the mechanical properties for the nano bainitic coatings.

### 3.2. XRD Analysis

The X-ray diffraction patterns of the coatings obtained after different isothermal times at 200 °C, 250 °C and 300 °C are shown in Figure 10a. It is very clear that phases in the coatings are mainly composed of α(α’) and γ phase, which represents the BF and RA respectively, and no diffraction peaks of carbides are seen. Figure 10b represents the elaborate observation of XRD patterns, which also exhibits that the peaks of ε($10\bar{1}0)$, ε(0002), and ε($10\bar{1}1)$ are slightly visible in the 200 and 250 °C coatings. The peaks of the hexagonal close-packed phase are weak as a relative fewer fraction of the ε-martensite phase in the coatings, but compared to the 300 °C coating, it is obvious that the ε-martensite phase indeed exists in 200 and 250 °C coatings. Talonen and Kundu have found that the bainitic transformation elastics could cause strain and stress fields. The strain and stress filed will stimulate the generation of the γ(111) stacking faults in the adjacent austenite. When the energy is accumulated to a specific value, the nucleation of the ε-martensite occurs and grows to be martensite. Further, the ε-martensite can only be observed in the coatings with longer time transformations, which also proved that the ε-martensite transformation was active by the strain and stress fields accumulation [44].

The volume fractions of each phase as a function of the isothermal treatment process by the metallography method based on the SEM and OM result, and direct comparison method based on the XRD results [45,46].

The carbon content of the austenite and bainite ferrite can be calculated by the relationship between the lattice parameter and chemical compositions, as following [28]:(1)$$a_{\gamma }=3.5780+0.033w_{C}+0.00095w_{Mn}-0.0002w_{Ni}+0.0006w_{Cr}+0.0054w_{Al}+0.0031w_{Mo}+0.0018w_{V}$$
(2)$$a_{\alpha }=2.8664+\frac{{\left(a_{Fe}-0.279x_{C}\right)}^{2}\left(a_{Fe}+2.496x_{C}\right)-a_{Fe}^{3}}{3a_{Fe}^{2}}-0.03x_{Si}+0.06x_{Mn}+0.07x_{Ni}+0.31x_{Mo}+0.05x_{Cr}+0.096x_{V}$$
where $a_{\gamma }$ is the lattice parameter of the austenite (Å), and wi is the content of element *i* (wt.%). xi is the mole fraction of elements *i*, and $a_{Fe}$ = 2.8664 Å is the lattice parameter of ferrite in pure iron. The least squares method was applied to calculate the accurate parameters based on the XRD data.

The calculated results based on the XRD and metallographic test concerning the microstructure as a function of transformation temperature are summarized in Table 4. It is found that the fraction of RA decreased with a decrease in isothermal temperature. The bainitic microstructure formed at low temperature contains retained austenite with film morphology. This is related to the high transformation ratios as a result of the high driving force at low transformation temperatures. The volume fraction of retained austenite with blocky (*V*γ_B_) and film (*V*γ_F_, including the island austenite) morphology is possible to estimate as a function of the isothermal temperatures following the reference [47]. The other significant effect of isothermal temperature on the microstructure is that the blocky morphology is predominant retained austenite at 300 °C, up to 24 vol.%. This phenomenon is also coinciding with the fraction of blocky austenite increasing with isothermal temperature, according to the previous analysis of the SEM and TEM results. The blocky retained austenite is less stable than either film and island-like austenite due to the lower carbon concentration.

The average carbon content decreases with the increasing isothermal temperature in the retained austenite, from 6.2 (at. %) of 200 °C to 4.7 (at. %) of 300 °C. The corresponding Ms temperatures of the retained austenite are −77.4 °C, −20 °C and 45.5 °C, respectively (calculated using JMatPro 7.0). Therefore, most of the retained austenite can maintain stability at ambient temperature. However, some martensite is observed in the 200 °C and 250 °C transformed coatings. The transformation is likely induced by the residual stress due to the rapid solidification process.

### 3.3. Mechanical Properties

Figure 11 shows the curves of Nano-hardness load-displacement, the distribution of corresponding nano-indentation hardness and elastic modulus in the middle region of the coatings. The middle region in the transverse cross-section of the coating was selected to eliminate the interference factors, such as dilution zone or difference of the dendrites which will be variation of the hardness across the cross-section [48]. There are no obviously metallurgical defects in the microstructure of the laser cladded bainitic coatings. The curves load-displacement of each group have different residual impression depths (h) shows that the hardness of the testing area are un-uniform. This is because the bainite ferrite has a relative higher hardness, greater than that of the retained austenite. The magnified view of curves at the loading stage show some pop-in events or ripples, and all of the curves is taken on the “soften zone”. This illustrates the plastic deformation of the blocky retained austenite.

The 200 °C transformed coatings has the highest average hardness (H, 8.39 GPa) and elastic modulus (E, 230), which are greater than that of the 250 °C (H, 8.11 GPa; E, 226) transformed coating. The 300 °C transformed coating has the minimum average H(6.26 GPa) and E (221), because it contained a significant retained austenite.

Compared with the microstructure morphology and distribution, most of the nano hardness distributes averagely in the inner dendrite area, and there are also some regions represents lower nano hardness, as the blue regions in the Figure 11. It indicates that the blocky retained austenite distributing at the interdendritic regions has a relatively lower hardness. The coatings have experienced a similar melting and solidification process, so the E of the interdendritic retained austenite in three isothermal coatings are also similar. Therefore, the nanoindentation hardness of the coatings is related to the scale of the bainite ferrite lath and the fractions of the bainite ferrite. The average width of the bainite ferrite lathes is increasing, and the decreasing bainite ferrite volume fractions are the main reasons for the decreasing nano hardness with the increasing isothermal temperatures.

The nano-indentation test results show that there are different nano hardness and elastic modulus across the dendrite region. The difference is caused by the microstructure difference between the interdendritic region and inner dendrite area. The nano scale bainite ferrite and film austenite distribute at the inner dendrite area. The blocky retained austenite and a small amount of martensite distribute at the interdendritic region. As mentioned above, residual stress field induced the martensite transformation of the unstable blocky retained austenite. The martensite has a high carbon content, which is hard and brittle. This is detrimental for the mechanical properties of the bainitic coatings. The un-uniformed microstructure and mechanical properties can be probed using micro-cantilevers method and micro-mechanical techniques [49,50]. Further studies of the nano bainitic coatings will be carried out by using such a novel method and techniques.

The tensile strength and strain of the coatings are tested at ambient temperature. The results are illustrated in Figure 12. The ultimate tensile strength is over 2098 MPa, and the total elongation is 3.9% for the coatings obtained at 200 °C. The ultimate tensile strength of the isothermal transformed coatings at the 250 °C decreases to 1887 MPa, and then 1448 MPa for the 300 °C coatings. On the contrary, the total elongation ($\varepsilon_{T}$) for the coatings isothermal transformed at 250 °C is 6.3% and then increases to 7.1% for the 300 °C. The tensile strength for the nano bainitic coatings is determined by three factors of the retained austenite volume fractions and morphologies, the scale of the bainitic lathes, and the dislocation density in the bainite and retained austenite [36,51]. It can be found that the tensile strength increases with decreasing isothermal temperature, because the strength of the bainitic coating is increases with the decreasing width of the bainitic lathes as the Hall-Petch equation. The higher dislocation density in the nano bainitic microstructures also enhances the strength of the coatings transformed at a relatively lower temperature.

Elongation of the nano bainitic coatings is determined by the volume fractions of the retained austenite and their mechanical stabilities [11,52]. The retained austenite, as a ductile phase, can enhance both of the strength and ductility as a consequence of the TRIP phenomenon under the application of the tensile strain. The mechanical stability of the retained austenite is another main factor for the elongations of the nano bainitic coatings. The mechanical stability of the RA is dependent on the chemical composition (carbon and other alloys) and its morphology (in blocky or films) [53,54]. According to the microstructure analysis and discussion in Section 3.1, the volume fraction of retained austenite decreases with isothermal temperature, and with increasing elongation.

SEM fractography of tensile tested specimens are shown in Figure 13. The appearance of the fractured surface of the 200 °C and 250 °C shows a quasi-cleavage fracture, tear ridges. Some ductile dimples are also visible. It is also can be noticed that the quasi-cleavage facets become wider and longer in coatings transformed at relative higher isothermal temperatures. These cleavages present the nano bainite sheaves of the microstructure of the growth morphologies. The tear ridges appear the morphology and distributions of the retained austenite. With the increasing transformation temperature, the fracture becomes more ductile, and the ductile dimples become a dominant morphology in the coatings transformed at 300 °C. The increasing ductility of the coatings is attributed to the increasing volume fractions in the nano bainite coatings with increasing isothermal temperatures. The cleavage fracture shows that the 200 °C and 250 °C transformed coatings are ductile and quasi cleavage fracture. The transformed samples at 300 °C show a signature of ductile fracture since the specimen contains a high volume fraction of retained austenite.

The tensile testes results show that the strength of the laser cladded nano bainitic coatings can be achieved at the same level as the bulk materials which experienced the homogenization and deformation processes [10]. But it can be seen that the ductility and the elongation are deteriorate compared with the bulk nano bainite steels. It may be caused by the inhomogeneous distributions of the chemical compositions for the laser cladded solidification processes. The micro-cracks can be observed in the coatings transformed at 300 °C, as shown in Figure 14. The micro-crack distributes at the prior interdendritic areas of the solidification microstructures, which has relatively lower stability compared with the inner retained austinites. It transformed to high carbon martensite. The crack occurs at the during the water-cooling process. The martensite can be observed under the TEM [55]. This caused the relative lower ductility of the laser cladded nano bainite.

## 4. Conclusions

The effect of dendrite segregation on the microstructure and mechanical properties of laser cladded bainitic was investigated. Three subsequent isothermal temperatures (200 °C, 250 °C, 300 °C) were applied to obtain the bainitic coatings with different size and morphology. The following conclusion can be drawn from this study:There are three type of retained austenite distribute in the bainitic coatings. The film RA and island-like RA distribute in lamellar ferrite while the blocky retained austenite distribute at the interdendritic region.The interdendritic blocky austenite is retained as a result the element segregation in this area, which also influences the bainitic growth morphologies in the different temperature transformed coating.The RA in the interdendritic region has relative lower stability, which can be transformed to martensite, which is detrimental for the mechanical properties.The nano indentation shows that the coatings obtained at 300 °C have a lower nano hardness (6.26 GPa) than those obtained at 200 °C (8.39 GPa) and 250 °C (8.11 GPa).The 200 °C specimens showed extremely high tensile strength (2098 MPa), but the elongation is lower than the 250 °C and 300 °C specimens.The fracture surfaces of the tensile specimens consisted of dimples, quasi-cleavage fracture, tear ridges, and characteristics of a mixed failure.

## Figures and Tables

**Figure 1 materials-13-03017-f001:**
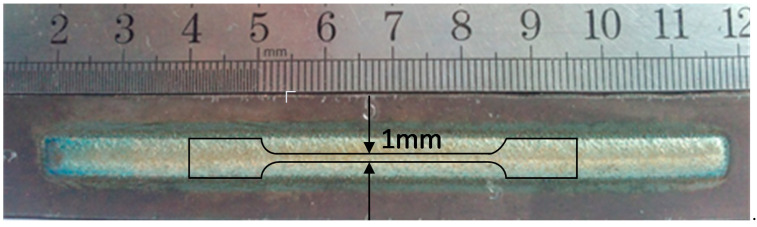
The appearance of the coating and the tensile test specimen.

**Figure 2 materials-13-03017-f002:**
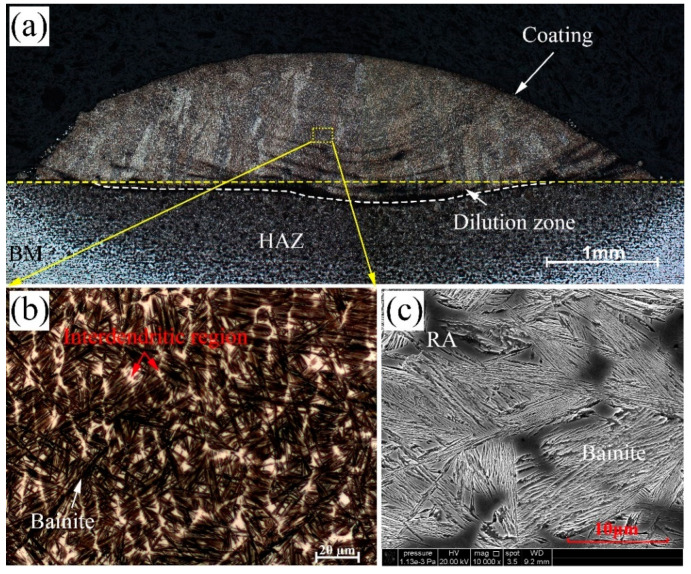
The profile of the laser cladded coating (**a**), and detailed images of the isothermal microstructures in the center region observed by OM (**b**) and SEM (**c**).

**Figure 3 materials-13-03017-f003:**
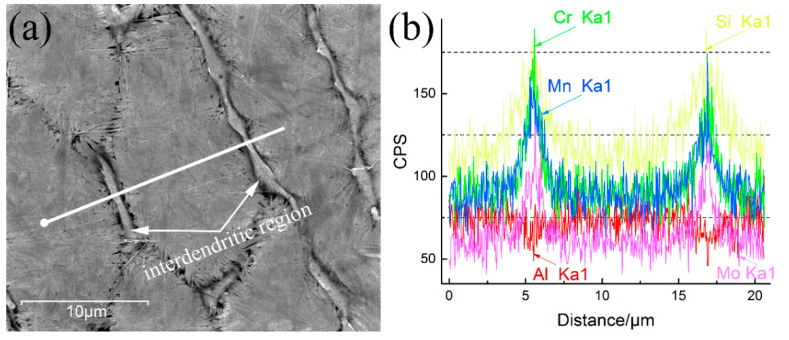
(**a**) A composite map of elements segregation across the interdendritic region. The white line represents the area incorporated into the vertically integrated line scanning shown in (**b**).

**Figure 4 materials-13-03017-f004:**
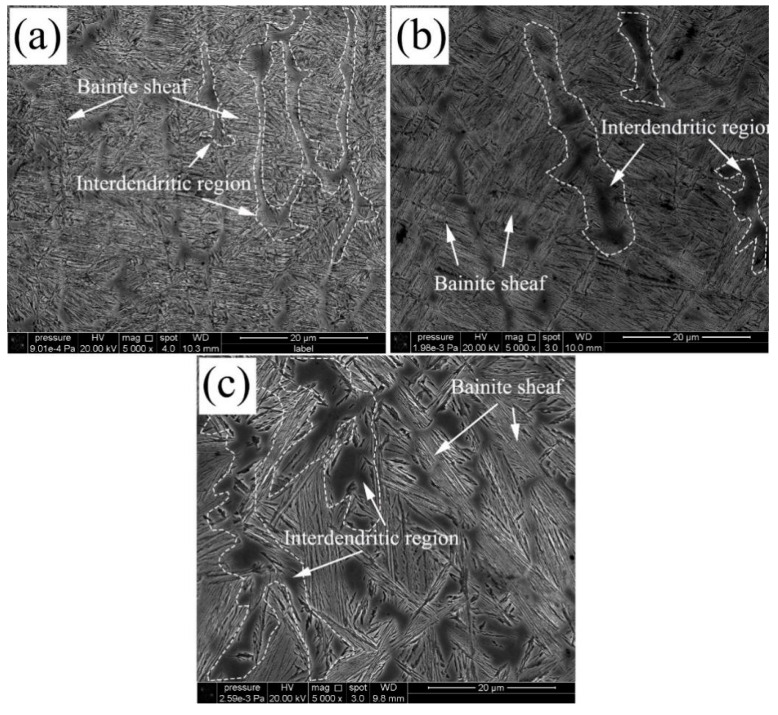
SEM morphologies of microstructure: (**a**) 200 °C transformed for 24 h, (**b**) 250 °C transformed for 16 h, (**c**) 300 °C transformed for 8 h.

**Figure 5 materials-13-03017-f005:**
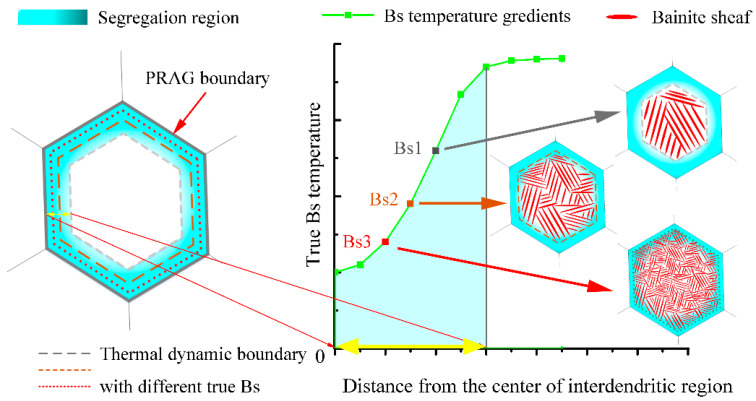
Schematic diagram of the interdendritic segregation effect on bainite and retained austenite distribution under different isothermal temperature (PRAG means the prior austenite grain).

**Figure 6 materials-13-03017-f006:**
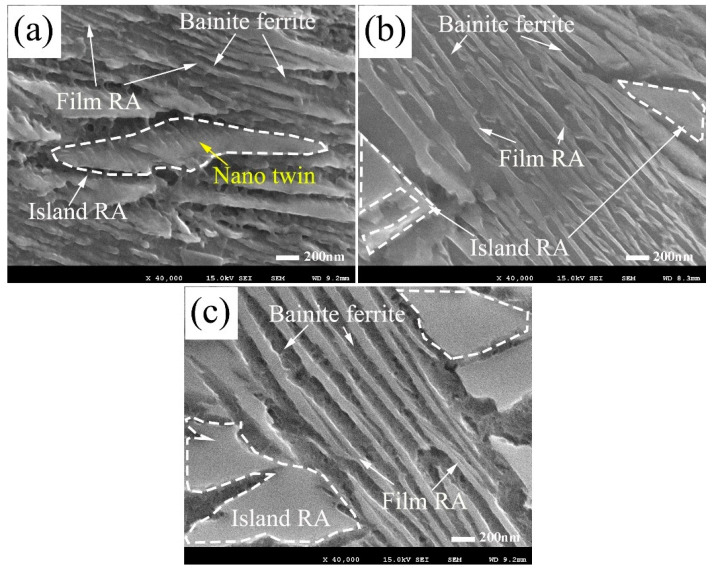
Magnified scanning electron micrographs of the inner dendrite regions in the coatings (**a**) obtained at 200 °C for 24 h, (**b**) obtained at 250 °C for 16 h, (**c**) obtained at 300 °C for 8 h.

**Figure 7 materials-13-03017-f007:**
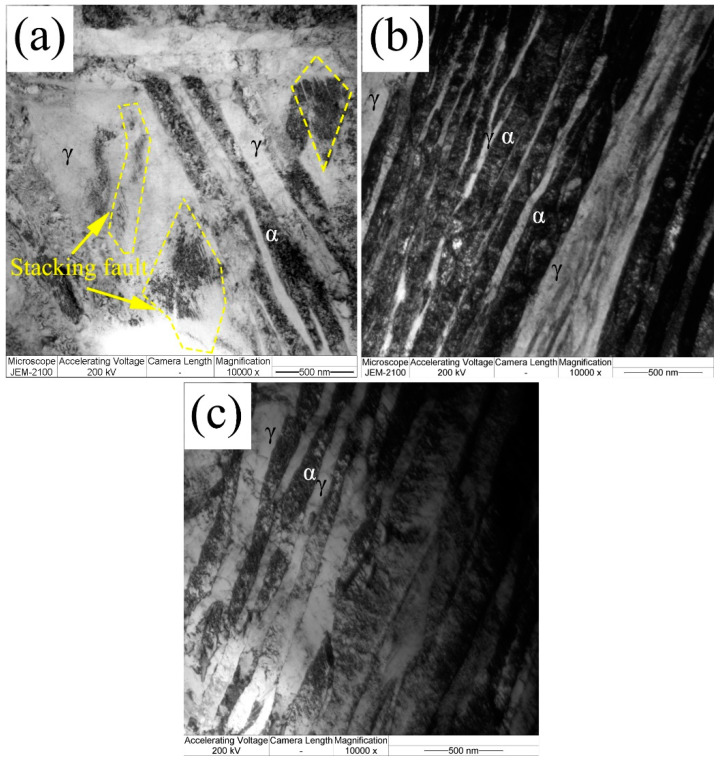
Typical TEM micrograph of the bainite transformed at (**a**) 200 °C, (**b**) 250 °C, and (**c**) 300 °C.

**Figure 8 materials-13-03017-f008:**
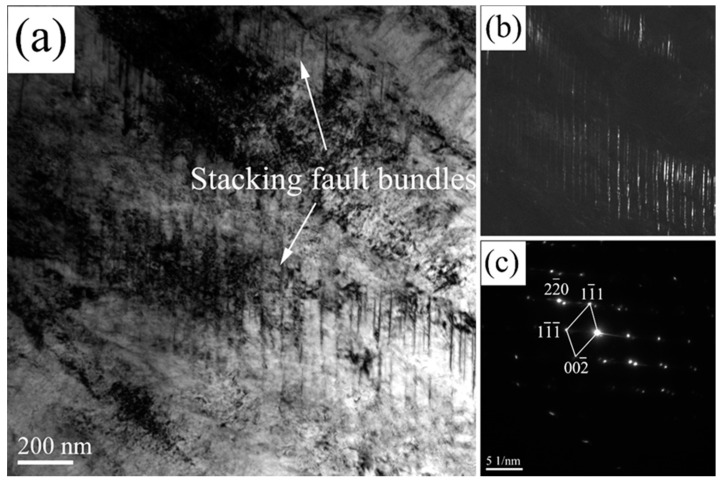
TEM images of the stacking fault bundles in the blocky RA of the 200 °C: (**a**) bright-field image, (**b**) dark-field images, and (**c**) SAED pattern.

**Figure 9 materials-13-03017-f009:**
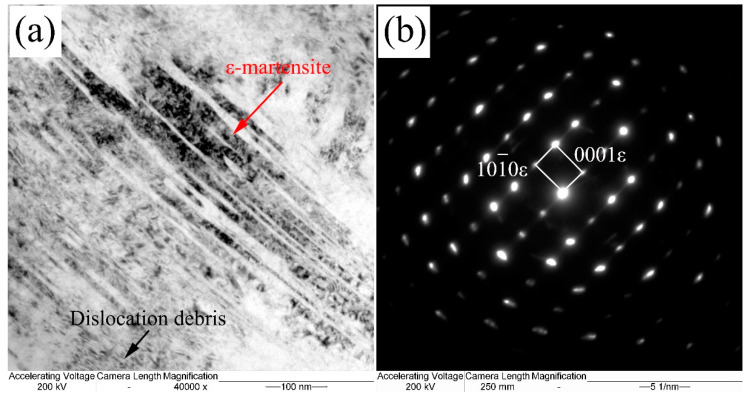
TEM morphology of ε-martensite in 200 °C transformed coating: (**a**) bright. field image, (**b**) SAED pattern.

**Figure 10 materials-13-03017-f010:**
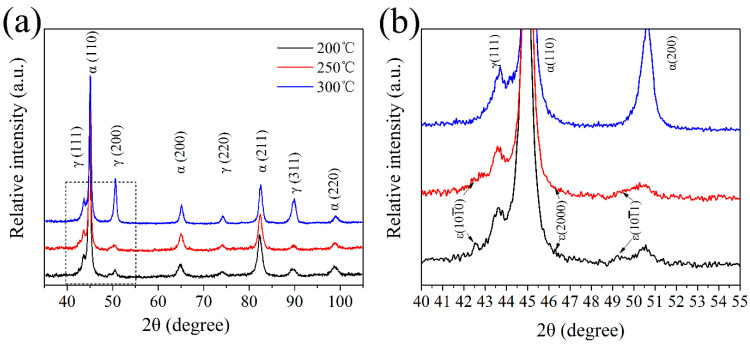
The XRD patterns of the isothermal transformed coatings (**a**), detailed of XRD patterns from 40° to 55° (**b**).

**Figure 11 materials-13-03017-f011:**
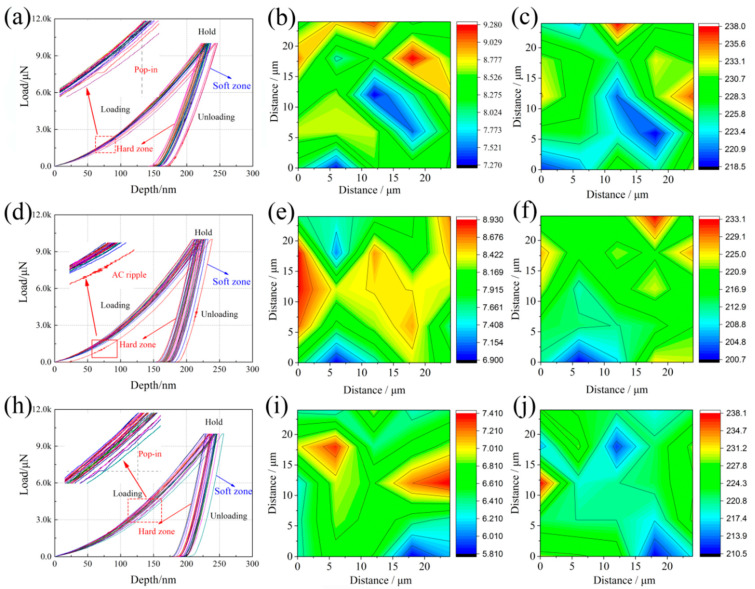
Nano-hardness load-displacement curves, the corresponding distributions of the nano-indentation hardness and elastic modulus in the nano bainitic coatings obtained at 200 °C (**a**–**c**); 250 °C (**d**–**f**); 300 °C (**h**–**j**).

**Figure 12 materials-13-03017-f012:**
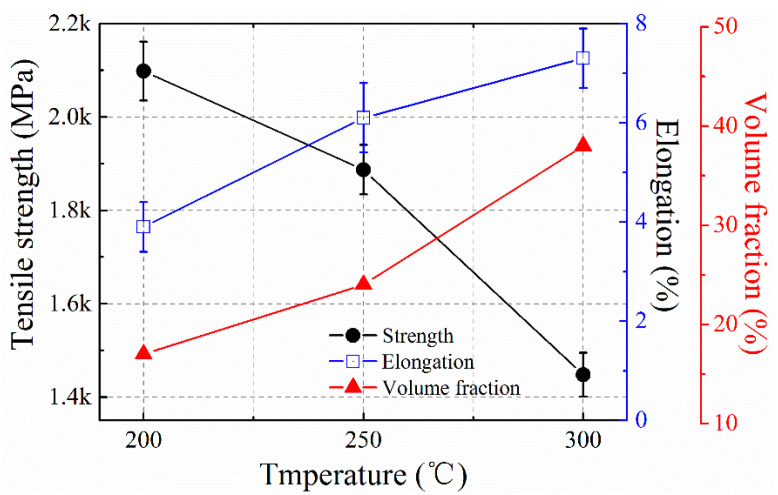
Mechanical properties of the laser cladded bainitic coatings.

**Figure 13 materials-13-03017-f013:**
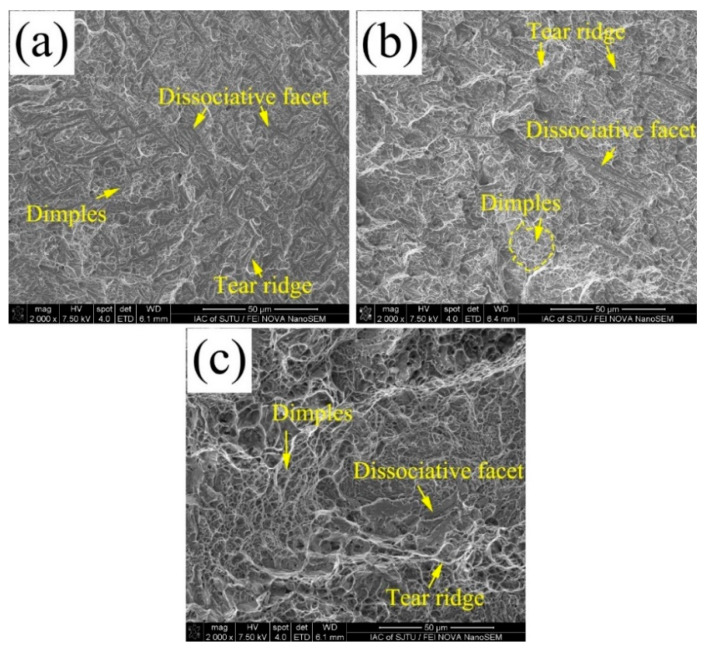
Typical tensile fractography of (**a**) the coatings obtained at 200 °C, and (**b**) the coatings obtained at 250 °C, and (**c**) the coatings obtained at 300 °C.

**Figure 14 materials-13-03017-f014:**
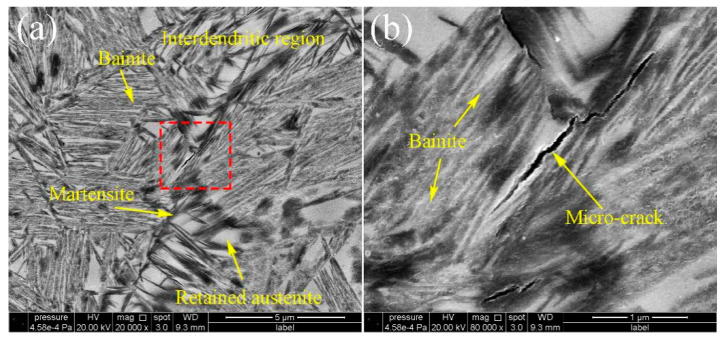
The micro-crack in the microstructure under the water cooling and the ε-martensite in the blocky austenite (**a**), the magnified view of the cracks (**b**).

**Table 1 materials-13-03017-t001:** Chemical composition of the powder and substrate (wt.%).

	C	Si	Mn	Cr	Mo	Co	Al	Fe
Powder	0.80	1.51	1.93	1.08	0.28	1.59	1.06	Bal.
Substrate	0.14	0.22	0.58	-	-	-	-	Bal.

**Table 2 materials-13-03017-t002:** Processing parameters for laser cladding.

Laser power (kW)	Scanning velocity (mm/s)	Spot diameter (mm)	Powder feed rate (g/min)	Shielding gas flow rate (L/min)	Tail shielding gas flow rate (L/min)
2.3	10	5	30	5	15
2.4	10	5	30	5	15
2.5	10	5	30	5	15

**Table 3 materials-13-03017-t003:** Heat treatment parameters for isothermal transformation.

Temperature (°C)	Isothermal Time (h)
300 °C	0.083/0.25/0.5/0.75/1/1.5/2/3/4/6/8/16
250 °C	0.083/0.25/1/2/3/4/8/12/16/20/24
200 °C	2/4/8/12/24/48/72/120/180/240

**Table 4 materials-13-03017-t004:** Carbon content and Volume fraction of retained austenite with different transformation temperature.

*T* (°C)	*V*γ (vol.%)	*V*γ_F_ (vol.%)	*V*γ_B_ (vol.%)	Cγ (at.%)
200	17	12	5	6.2
250	24	15	9	5.5
300	38	14	24	4.7

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
