# Peer review of "The Effects of Micro-Segregation on Isothermal Transformed Nano Bainitic Microstructure and Mechanical Properties in Laser Cladded Coatings"

_materials, 2020, doi:10.3390/ma13133017_

Round 1

Reviewer 1 Report

Please find my comments in the attached pdf.

Reviewer 2 Report

The authors provide a paper dealing with the effects of micro-segregation on isothermal transformed nano bainitic microstructure and the analysis of mechanical properties. This is an interesting study with a complete characterization and nice results. I however have some comments in order to improve the paper.

  • The analysis of mechanical properties, especially the nanoindentation, is not so clear to me. Figure 11 reports all the nanoindentation curves and it not clear to me why. I would rather more interesting to an analysis of the elastic modulus (to be added) and some improvement for the hardness. Specifically, I don’t understand the 3D plot, actually I think that it should be only a 2D plot. What is the penetration depth in which E and H are taken? Do the authors detect any effect to the loading rate?
  • I think that the analysis of the mechanic properties can be improved by commented on residual stress and fracture toughness. Specifically, the authors must be aware that there exist technqiues to probe the local residual stress and fracture toughness. As reported in doi.org/10.1016/j.matdes.2019.107762 and doi.org/10.1016/j.matdes.2016.06.003. The authors are invited to comment on these papers, and I think that those techniques are very interested in this work in which a non-uniform bainitic microstructure is investigated.
  • The experimental part of nanoindentation need to be improved describing the loading rate, calibration of the area function etc. Moreover, I think it can be nice to add some experiment in continuous stiffness measurement configuration (CSM) providing the E and H as a function of the indentation depth.

Round 2

Reviewer 1 Report

The authors have addressed all of my comments in sufficient detail. I have no further comments/feedback.

Reviewer 2 Report

-